# Inference-Time Reward-Guided Langevin Refinement for Diffusion Models

## Abstract

We study test-time alignment of pretrained diffusion models to task-specific rewards for ODE-based samplers. Starting from a DPO-inspired, energy-based distribution, we introduce an inference-time Langevin refinement that interleaves standard deterministic denoising updates with reward-guided corrections applied in image/latent space. The procedure is plug-and-play for common ODE solvers, requires no additional training or data, and works with arbitrary differentiable reward models—including human-preference reward models, aesthetic or safety scores, and CLIP-based rewards. Empirically, across multiple datasets and rewards, the method consistently increases reward at test time. The result is a lightweight "align-as-you-sample" approach that turns pretrained diffusion models into reward-seeking generators to increase quality without finetuning data, extra training, or architectural changes.

## 1 Introduction

In recent years, denoising diffusion models have rapidly emerged as the most standard method for high-fidelity image generation, driving remarkable progress in both academic research and industrial applications (Ho et al., 2020; Dhariwal & Nichol, 2021; Rombach et al., 2022). Their ascendancy is underpinned by a framework that couples exceptional practical performance with a remarkably solid theoretical foundation. Unlike alternative generative paradigms, diffusion models are grounded in a well-established formulation based on Stochastic Differential Equations (SDEs) (Song et al., 2020b). This perspective provides a unified and principled mathematical framework that elegantly connects the forward noising process and the reverse generative process—together with its deterministic probability-flow ODE counterpart (Song et al., 2020b) and efficient ODE samplers such as DDIM (Song et al., 2020a)—offering guarantees and insights that are often less explicit in other models. The combination of this theoretical elegance with the ability to produce state-of-the-art results in terms of sample quality and diversity (Karras et al., 2022; Lu et al., 2022; Dhariwal & Nichol, 2021) makes diffusion the most popular approach for image generation.

A key frontier in advancing generative models is aligning their outputs with nuanced human preferences, moving beyond mere visual fidelity towards attributes like aesthetic quality, composition, and ethical compliance. To this end, a growing body of research focuses on fine-tuning generative models using human feedback. Existing approaches can be broadly categorized along two axes: the form of feedback and the training paradigm. Regarding feedback, methods leverage either direct human-provided data, such as pairwise comparisons (Wallace et al., 2024), or a learned reward function that encapsulates human preferences (Fan et al., 2023). In terms of the training paradigm, some techniques involve a full or partial re-training of the model, often framed as reinforcement learning (RL) where the denoising process is optimized to maximize a reward signal (Fan et al., 2023; Clark et al., 2023). Alternatively, other methods employ distillation to transfer knowledge from a reward-guided process into the model parameters (Salimans & Ho, 2022). Conversely, a distinct line of work operates at *inference-time*, steering pre-trained models without updating their weights by using guidance signals derived from classifiers or reward models (Dhariwal & Nichol, 2021; Ho & Salimans, 2022). While effective, these inference-time methods often face a trade-off between control strength and sample quality.

In this work, we introduce a *training-free, inference-time* framework that aligns deterministic ODE-based diffusion samplers to an arbitrary *differentiable* reward function. At each step, we perform

a short *Langevin refinement* targeting a *pulled-back* energy tilt, a concept rooted in Energy-Based Models (LeCun et al., 2006; Du et al., 2023), and then apply the deterministic denoiser to push the refined state to the next step. This interleaving yields a simple, compositional guarantee: the next-step distribution realizes the intended exponential tilt by the reward, and iterating across steps recovers the DPO-style energy-based distribution target (Wallace et al., 2024), drawing a formal connection to the principles of Direct Preference Optimization (Rafailov et al., 2023). Our method is plug-and-play, architecture-agnostic, and compatible with diverse rewards (Aesthetics (Fan et al., 2023), NIMA (Talebi & Milanfar, 2018), classifier-based class guidance (Dhariwal & Nichol, 2021)), providing practical preference alignment entirely at inference time, in contrast to methods requiring fine-tuning (Fan et al., 2023; Clark et al., 2023) or reward model training (Rafailov et al., 2023; Wallace et al., 2024).

To summarize, the main contributions of our paper are as follows:

**Training-free, inference-time framework.** We introduce a reward-guided, *training-free* inference-time finetuning framework for *deterministic* ODE-based diffusion samplers, interleaving short Langevin refinement with the model's deterministic denoiser.

**Theoretical guarantees.** We provide rigorous proofs showing that our layer-wise refinement and deterministic push yield the intended exponential-tilted distribution at each layer, culminating in the DPO-style tilt at the output, thereby establishing the framework's correctness.

**Empirical effectiveness.** Extensive experiments demonstrate that our method consistently increases the *expected reward* across datasets and reward models; moreover, using *classifier logits* as the reward enables *class-guided* image generation at inference time without parameter updates.

## 2 PRELIMINARIES

### 2.1 THE FORWARD AND BACKWARD PROCESS FOR DIFFUSION

We consider a forward noising Itô SDE on $\mathbb{R}^d$ over a finite horizon $[0, T]$:

$$dx_t = f(x_t, t)\, dt + g(t)\, dW_t, \qquad t \in [0, T], \quad x_0 \sim p_0 \equiv p_{\text{data}}, \tag{2.1}$$

where $f(\cdot, t)$ is a drift field, $g(t) > 0$ a scalar (or isotropic) diffusion schedule, and $W_t$ a standard Wiener process. Let $p_t$ denote the marginal density of $x_t$ induced by equation 2.1.

The *reverse-time (backward) SDE* that transports $x_T \sim p_T$ back to $x_0 \sim p_0$ reads

$$dx_t = \Big(f(x_t, t) - g(t)^2\, \nabla_x \log p_t(x_t)\Big) dt + g(t)\, d\bar{W}_t, \qquad t : T \to 0, \tag{2.2}$$

where $\bar{W}_t$ is a standard Wiener process in reverse time and $\nabla_x \log p_t$ is the (unknown) score.

A deterministic counterpart, the *probability flow (backward) ODE*, shares the same time marginals when the score is exact:

$$\frac{dx_t}{dt} = f(x_t, t) - \frac{1}{2} g(t)^2\, \nabla_x \log p_t(x_t), \qquad t : T \to 0. \tag{2.3}$$

In practice, a pretrained score model $s_\theta(x, t) \approx \nabla_x \log p_t(x)$ replaces the unknown score in equation 2.2 and equation 2.3, and numerical discretizations of equation 2.3 yield deterministic ODE-based denoising samplers (e.g., DDIM/EDM-style integrators (Song et al., 2020a; Karras et al., 2022)).

### 2.2 LANGEVIN DYNAMICS AS AN SDE AND ITS DISCRETIZATION

Given a target density $\pi(x) \propto e^{-U(x)}$ with energy $U : \mathbb{R}^d \to \mathbb{R}$, the *overdamped Langevin SDE* (unit temperature) is

$$dx_\tau = -\nabla U(x_\tau)\, d\tau + \sqrt{2}\, dB_\tau = \nabla \log \pi(x_\tau)\, d\tau + \sqrt{2}\, dB_\tau, \tag{2.4}$$

where $B_\tau$ is standard Brownian motion. The associated Fokker–Planck equation $\partial_\tau q_\tau = -\nabla \cdot \big(q_\tau \nabla \log \pi\big) + \Delta q_\tau$ admits $\pi$ as a stationary distribution under mild conditions (e.g., confining $U$) (Pavliotis, 2014). A basic time-discretization is the *Unadjusted Langevin Algorithm (ULA)* (Robert & Tweedie, 1996; Parisi & Yongshi, 1980):

$$x \leftarrow x + \alpha\, \nabla \log \pi(x) + \sqrt{2\alpha}\, \xi, \qquad \xi \sim \mathcal{N}(0, I),\ \alpha > 0, \tag{2.5}$$

which converges to $\pi$ as $\alpha \to 0$ with sufficiently many iterations (or exactly with a Metropolis correction).

## 2.3 RLHF-STYLE OBJECTIVE AND DPO: THE INDUCED ENERGY-TILTED OPTIMUM

Let $p_\theta(x_0)$ be a parametric conditional-free (unconditional) generator on images $x_0$, and let $p_{\text{ref}}(x_0)$ be a fixed reference distribution (e.g., a pretrained model). RLHF aims to maximize a latent reward $r(x_0)$ while regularizing the KL-divergence to $p_{\text{ref}}$ (Ziegler et al., 2019; Stiennon et al., 2020):

$$\max_{p_\theta} \ \mathbb{E}_{x_0 \sim p_\theta(x_0)}\big[ r(x_0) \big] \ - \ \beta \, \mathrm{D}_{\text{KL}}\big[ p_\theta(x_0) \,\|\, p_{\text{ref}}(x_0) \big], \qquad \beta > 0. \tag{2.6}$$

For the objective equation 2.6, the (unique) global optimum takes the *energy-tilted* closed form:

$$p_\theta^\star(x_0) \ = \ \frac{1}{Z} \, p_{\text{ref}}(x_0) \, \exp\big(r(x_0)/\beta\big), \quad Z \ = \ \int p_{\text{ref}}(x_0) \, \exp\big(r(x_0)/\beta\big) \, dx_0. \tag{2.7}$$

**Derivation**   Introducing a Lagrange multiplier $\lambda$ for normalization and maximizing the functional

$$\mathcal{L}[p] = \int p(x_0) \, r(x_0) \, dx_0 - \beta \int p(x_0) \log \frac{p(x_0)}{p_{\text{ref}}(x_0)} dx_0 - \lambda \left( \int p(x_0) dx_0 - 1 \right) \tag{2.8}$$

w.r.t. $p$ yields

$$\frac{\delta \mathcal{L}}{\delta p} = r(x_0) - \beta \big( \log p(x_0) - \log p_{\text{ref}}(x_0) + 1 \big) - \lambda = 0 \tag{2.9}$$

hence $\log p(x_0) = \log p_{\text{ref}}(x_0) + r(x_0)/\beta - C$. Exponentiating and normalizing gives equation 2.7.

**Energy-based view.**   Eq. equation 2.7 is an *energy-based model* (EBM) with energy $E(x_0) = -\log p_{\text{ref}}(x_0) - r(x_0)/\beta$. Thus, optimizing equation 2.6 is equivalent to sampling from the EBM family $p(x_0) \propto p_{\text{ref}}(x_0) \exp(r(x_0)/\beta)$.

## 2.4 ODE-BASED DIFFUSION: DETERMINISTIC DENOISING

We adopt a deterministic ODE-based sampler (e.g., a DDIM/EDM-style integrator of the probability flow ODE), which produces a discrete-time denoising trajectory

$$x_{t-1} \ = \ f_t(x_t), \qquad t \in \{T, T-1, \ldots, 1\}. \tag{2.10}$$

Let $p_t$ denote the model-induced marginal of $x_t$ at layer $t$. As $f_t : \mathbb{R}^d \to \mathbb{R}^d$ is (differentiable and locally invertible on the support), the layer-wise densities satisfy the change-of-variables identity

$$p_t(x) \ = \ p_{t-1}\big(f_t(x)\big) \, \big| \det \nabla f_t(x) \big|. \tag{2.11}$$

Hence, once $x_T \sim p_T$ (often standard Gaussian), the entire path $\{x_t\}$ is *deterministically* determined by equation 2.10; i.e., conditioning on $x_T$ fixes all subsequent $x_{T-1}, \ldots, x_0$.

## 3 METHOD

**Ultimate target**   Motivated by the DPO closed form, we aim to tilt the output distribution at $t=0$ to the energy-based model

$$p_0^{(\lambda)}(x) \ \propto \ p_0(x) \, \exp\big(\lambda \, r(x)\big), \tag{3.1}$$

where $r : \mathcal{X} \to \mathbb{R}$ is a (differentiable) reward and $\lambda > 0$ is the global reward strength.

**Step-wise targets**   To reach equation 3.1 progressively, we introduce a decreasing schedule

$$\lambda_T = 0, \qquad \lambda_0 = \lambda, \qquad \lambda_{t-1} \geq \lambda_t \ \text{for} \ t = 1, \ldots, T, \tag{3.2}$$

and consider the layer-$t$ tilt

$$p_t^{(\lambda_t)}(x) \ \propto \ p_t(x) \, \exp\big(\lambda_t \, r(x)\big). \tag{3.3}$$

Given the current $x_t' \sim p_t^{(\lambda_t)}$, we will produce $x_{t-1}' \sim p_{t-1}^{(\lambda_{t-1})}$ by two steps:

**Step 1**   From $x'_t$ at time $t$, we apply a few Langevin steps targeting

$$\pi_t(x) \;\propto\; p_t(x)\,\exp\big(\lambda_{t-1}\,r\big(f_t(x)\big)\big), \tag{3.4}$$

to obtain a refined sample

$$x_t^{\text{new}} \;\sim\; p_t(x)\,\exp\big(\lambda_{t-1}\,r(f_t(x))\big). \tag{3.5}$$

The continuous-time Langevin SDE targeting $\pi_t$ is:

$$dX_\tau \;=\; \nabla_x \log \pi_t(X_\tau)\,d\tau \;+\; \sqrt{2}\,dB_\tau. \tag{3.6}$$

Its log-gradient is:

$$\nabla_x \log \pi_t(x) \;=\; s_\theta(x,t) \;+\; \lambda_{t-1}\,\nabla_x r\big(f_t(x)\big). \tag{3.7}$$

Starting from the current layer state, we initialize:

$$x^{(0)} \;=\; x'_t. \tag{3.8}$$

We then run $K_t$ steps of ULA with step size $\alpha_t > 0$:

$$x^{(k+1)} \;=\; x^{(k)} \;+\; \alpha_t\Big[\,s_\theta\big(x^{(k)},t\big) \;+\; \lambda_{t-1}\,\nabla_x r\big(f_t(x^{(k)})\big)\Big] \;+\; \sqrt{2\alpha_t}\,\xi^{(k)}. \tag{3.9}$$

where the Gaussian noise is:

$$\xi^{(k)} \sim \mathcal{N}(0, I). \tag{3.10}$$

On very low-noise layers, we may use the noiseless variant:

$$x^{(k+1)} \;=\; x^{(k)} \;+\; \alpha_t\Big[\,s_\theta\big(x^{(k)},t\big) \;+\; \lambda_{t-1}\,\nabla_x r\big(f_t(x^{(k)})\big)\Big]. \tag{3.11}$$

After $K_t$ iterations, we set:

$$x_t^{\text{new}} \;:=\; x^{(K_t)}. \tag{3.12}$$

When we take $\alpha_t$ sufficiently small and $K_t$ sufficiently large, we would have $x_t^{\text{new}}$ satisfy equation 3.5.

**Step 2**   Given $x_t^{\text{new}}$ that satisfies equation 3.5. We then push the refined state through the deterministic denoiser:

$$x'_{t-1} \;=\; f_t\big(x_t^{\text{new}}\big), \tag{3.13}$$

which (by change of variables) yields the desired tilt at the next layer:

$$x'_{t-1} \;\sim\; p_{t-1}(x)\,\exp\big(\lambda_{t-1}\,r(x)\big). \tag{3.14}$$

*Proof.* By equation 3.5, $X' \equiv x_t^{\text{new}}$ has density $\tilde{\pi}_t(x) \propto p_t(x)\exp(\lambda_{t-1}r(f_t(x)))$. Let $Y = f_t(X')$. For any measurable $A \subset \mathbb{R}^d$,

$$\mathbb{P}(Y \in A) = \int \mathbf{1}_{\{f_t(x) \in A\}}\, p_t(x)\, e^{\lambda_{t-1} r(f_t(x))}\, dx. \tag{3.15}$$

Using the change-of-variables relation $p_t(x) = p_{t-1}(f_t(x))\,|\det \nabla f_t(x)|$ and $y = f_t(x)$,

$$\mathbb{P}(Y \in A) = \frac{1}{Z}\int_A p_{t-1}(y)\, e^{\lambda_{t-1} r(y)}\, dy, \tag{3.16}$$

for a normalization constant $Z$. Hence $Y$ has density $\tilde{\pi}_{t-1}(y) \propto p_{t-1}(y)\exp(\lambda_{t-1}r(y))$, i.e., the distribution in equation 3.14. $\qquad\square$

**Induction.**   Assuming $x'_t \sim p_t(x)\exp(\lambda_t r(x))$ holds at step $t$, applying Step 1 and Step 2 produces $x'_{t-1} \sim p_{t-1}(x)\exp(\lambda_{t-1}r(x))$. Iterating for $t = T, \ldots, 1$ establishes

$$x'_0 \;\sim\; p_0(x)\,\exp\big(\lambda\,r(x)\big),$$

which coincides with the DPO-optimal energy-based distribution equation 3.1.

Based on the deduction above, we get the algorithm below as a summary of our method.

---

**Algorithm 1** Reward-Guided Langevin Refinement at inference

---

**Require:** Score $s_\theta(x, t)$; deterministic denoiser $x_{t-1} = f_t(x_t)$; differentiable reward $r : \mathcal{X} \to \mathbb{R}$;
schedule $\{\lambda_t\}_{t=0}^T$ with $\lambda_T=0$, $\lambda_0=\lambda$; Langevin step sizes $\{\alpha_t\}$; iterations $\{K_t\}$;
  1: Sample $x'_T \sim \mathcal{N}(0, I)$                                            ▷ initialize the ODE-based sampler
  2: **for** $t = T, T-1, \ldots, 1$ **do**
  3:     $z \leftarrow x'_t$                                                         ▷ current state at time $t$
  4:     $x^{(0)} \leftarrow z$
  5:     **for** $k = 0$ **to** $K_t - 1$ **do**
  6:         $g^{(k)} \leftarrow s_\theta\big(x^{(k)}, t\big) + \lambda_{t-1} \nabla_x r\big(f_t(x^{(k)})\big)$
  7:         Sample $\xi^{(k)} \sim \mathcal{N}(0, I)$
  8:         $x^{(k+1)} \leftarrow x^{(k)} + \alpha_t\, g^{(k)} + \sqrt{2\alpha_t}\, \xi^{(k)}$
  9:     $z \leftarrow x^{(K_t)}$                                                    ▷ $x_t^{\mathrm{new}}$ as in equation 3.5
 10:     $x'_{t-1} \leftarrow f_t(z)$
 11: **return** $x'_0$                                                              ▷ final sample with target equation 3.1

---

## 4 EXPERIMENT

### 4.1 EXPERIMENTAL SETUP

**Backbone & sampler.** We use a pretrained diffusion backbone with a deterministic ODE-based sampler (e.g., DDIM/EDM probability-flow), discretized into $T=50$ steps from $t=T$ to $t=0$.

**Datasets.** We evaluate on four unconditional datasets at $256\times256$: CelebA-HQ(Karras et al., 2017), AFHQ (Cats/Dogs/Wild)(Choi et al., 2020), LSUN Bedroom(Yu et al., 2015), and ImageNet(Deng et al., 2009). All datasets are used with their standard training splits and standard preprocessing for the diffusion backbone.

**Reward models (differentiable).** We consider two differentiable rewards:

- **Aesthetic** (LAION aesthetic predictor v2): CLIP image encoder (ViT-L/14) (Radford et al., 2021) followed by a linear regression head to predict a scalar aesthetic score (Schuhmann et al., 2021). Images are resized to $224\times224$ with CLIP-standard normalization before scoring; the resize is differentiable and gradients flow back to the original $256\times256$ image.
- **NIMA** (Neural Image Assessment): We use the normal NIMA modelinputs are resized/cropped to $224\times224$ by bilinear interpolation.

Both rewards are used as frozen across all datasets to ensure fairness and reproducibility. For stability when combining rewards, both rewards output a score in the range $[0, 10]$.

**Reward schedule and guidance.** We follow the step-wise targets in Eqs. (3.2) and (3.4): $\lambda_T=0$, $\lambda_0=\lambda$, and $\lambda_{t-1} \geq \lambda_t$. Unless otherwise noted, we set $\lambda_{t-1} = \lambda \cdot w_t$ with a cosine decay

$$w_t = \tfrac{1}{2}\Big(1 + \cos\Big(\pi\, \tfrac{t-1}{T-1}\Big)\Big), \qquad t = 1, \ldots, T, \tag{4.1}$$

so that $w_1 = 1$ and $w_T = 0$.

**Langevin refinement.** At each time step $t$, we run $K_t \in \{2, 5, 10, 20\}$ steps of ULA:

$$x^{(k+1)} = x^{(k)} + \alpha_t\Big[ s_\theta(x^{(k)}, t) + \lambda_{t-1} \nabla_x r\big(f_t(x^{(k)})\big) \Big] + \sqrt{2\alpha_t}\, \xi^{(k)}. \tag{4.2}$$

to show the expected reward of images generated by our method under varying numbers of Langevin refinement steps.

**Implementation details.** Images are generated at $256\times256$ and fed to reward models after a differentiable resize to $224\times224$. We ensure gradients are enabled end-to-end during refinement (no accidental `no_grad` or score detachment). For each dataset, we generated 2000 images to calculate the average reward.

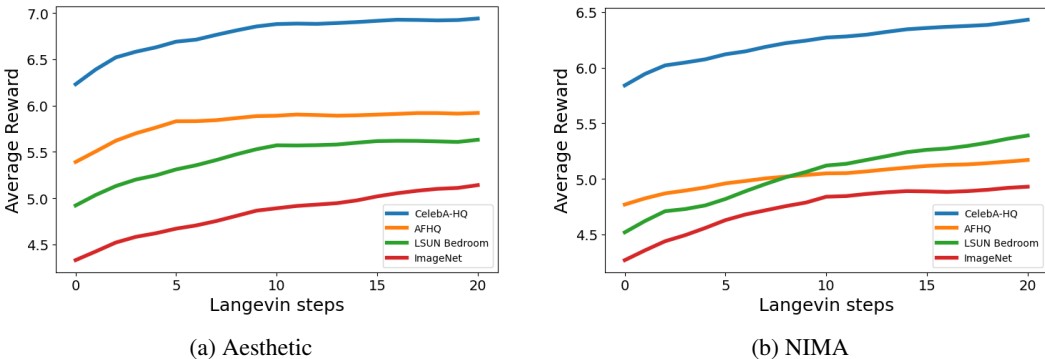

|                | (a) Aesthetic | (b) NIMA |

Figure 1: The average reward using our method under different Langevin steps. Showing significant increase when using more Langevin steps.

Table 1: Average reward under Aesthetic (left) and NIMA (right) on four datasets with different Langevin steps. Higher is better.

| Method | Aesthetic | | | | NIMA | | | |
|---|---|---|---|---|---|---|---|---|
| | CelebA-HQ | AFHQ | LSUN Bedroom | ImageNet | CelebA-HQ | AFHQ | LSUN Bedroom | ImageNet |
| Original | 6.23 | 5.39 | 4.92 | 4.33 | 5.84 | 4.77 | 4.52 | 4.27 |
| Langevin step=2 | 6.52 | 5.62 | 5.13 | 4.52 | 6.02 | 4.87 | 4.71 | 4.44 |
| Langevin step=5 | 6.69 | 5.83 | 5.31 | 4.67 | 6.12 | 4.96 | 4.82 | 4.63 |
| Langevin step=10 | 6.88 | 5.89 | 5.57 | 4.89 | 6.27 | 5.05 | 5.12 | 4.84 |
| Langevin step=20 | 6.94 | 5.92 | 5.63 | 5.14 | 6.43 | 5.17 | 5.39 | 4.93 |

## 4.2 MAIN RESULTS

Across all four $256 \times 256$ datasets and both differentiable rewards (Aesthetic and NIMA), our method monotonically increases the *expected reward* as the number of Langevin refinement steps grows. Crucially, **only 2 steps already yield a clear improvement**: Aesthetic increases by **+0.29** on CelebA-HQ (6.23→6.52), **+0.23** on AFHQ (5.39→5.62), **+0.21** on LSUN (4.92→5.13), and **+0.19** on ImageNet (4.33→4.52); NIMA increases by **+0.18** on CelebA-HQ (5.84→6.02), **+0.10** on AFHQ (4.77→4.87), **+0.19** on LSUN (4.52→4.71), and **+0.17** on ImageNet (4.27→4.44).

When we set **Langevin_steps = 20**, the gains become *very pronounced*: Aesthetic improves by **+0.81** on ImageNet (4.33→5.14), **+0.71** on LSUN (4.92→5.63), **+0.71** on CelebA-HQ (6.23→6.94), and **+0.53** on AFHQ (5.39→5.92); NIMA improves by **+0.87** on LSUN (4.52→5.39), **+0.66** on ImageNet (4.27→4.93), **+0.59** on CelebA-HQ (5.84→6.43), and **+0.40** on AFHQ (4.77→5.17).

Figure 1 shows the average reward of generated images would increase significantly as the Langevin steps increase. Overall, our inference-time refinement *significantly* boosts the expected reward with consistent trends across datasets.

## 4.3 CLASS GUIDANCE WITH OUR METHOD

Without modifying diffusion model parameters, our method can also provide class guidance to generate specific type of image at inference time using a frozen image classifier as the reward source. Concretely, we take the classifier's evidence for a target class (e.g., the class logit or log-probability) as a differentiable reward and plug it into our layer-wise refinement and deterministic push pipeline. This yields class-selective guidance that biases the generated samples toward the desired category while preserving the stability and controllability of the ODE-based sampler. In practice, we simply pass the generated image through a differentiable bilinear resize to the classifier's native resolution and apply the classifier's standard normalization before computing the scalar reward; no retraining is required. Also, choosing classifier as reward guidance using our method not only leads to class-

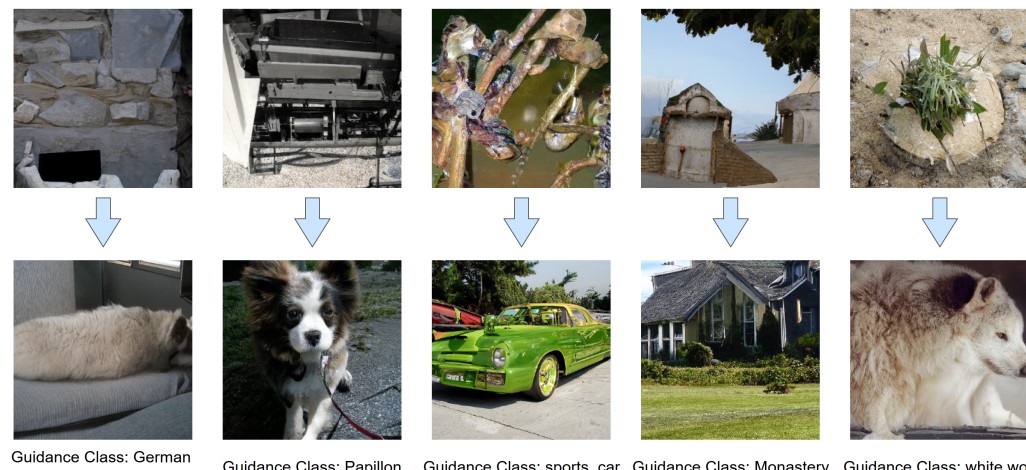

Figure 2: Use classifier as reward to generate class-specific images. This process not only generates a specific kind of picture but also increases the image quality.

specific image generation, but also increase the quality of generated images. Figure 2(on Imagenet $256 \times 256$) illustrates the effectiveness of our approach.

## 5 RELATED WORK

**Diffusion Models and Deterministic ODE Sampling**  The pursuit of efficient sampling has been a central theme in diffusion model research. A pivotal advancement came from the formulation of diffusion models as solutions to Stochastic Differential Equations (SDEs), which naturally admit a deterministic counterpart known as the Probability Flow ODE (Song et al., 2020b). This formulation enables deterministic sampling, where the generative process is defined by a unique trajectory, allowing for meaningful latent interpolation and exact reproducibility. Song et al. introduced the Denoising Diffusion Implicit Model (DDIM) sampler, a first-order solver that dramatically accelerates sampling while maintaining quality (Song et al., 2020a). Subsequent work has focused on developing more sophisticated ODE solvers. For instance, Karras et al. (2022) empirically elucidated design choices for ODE-based sampling, leading to robust practices. Lu et al. (2022) proposed DPM-Solver, a higher-order solver tailored to the specific structure of the diffusion ODE, achieving high-fidelity generation in as few as 10 steps. Beyond designing solvers, another line of work, such as progressive distillation (Salimans & Ho, 2022), aims to distill the behavior of a complex ODE solver into a more efficient model, pushing the limits of sampling speed further. These efforts collectively establish deterministic ODE sampling as a powerful and efficient paradigm for diffusion models.

**Guidance for Diffusion Model**  Controlling the outputs of diffusion models to align with human preferences, a challenge often addressed by Reinforcement Learning from Human Feedback (RLHF) in language models, has seen parallel advancements in the image domain. Early work focused on inference-time guidance. Dhariwal & Nichol (2021) pioneered classifier guidance, which uses the gradients from a pre-trained classifier to steer the sampling process towards a desired attribute. This was later superseded by classifier-free guidance, a more robust and widely adopted technique that leverages a jointly trained conditional and unconditional model to achieve control without an auxiliary classifier (Ho & Salimans, 2022). More recently, research has explored directly incorporating human feedback into the model training loop, mirroring RLHF paradigms. Some approaches fine-tune diffusion models directly using reward gradients (Clark et al., 2023), while others, like DPOK, formalize denoising as a sequential decision-making problem and apply policy gradient algorithms (Fan et al., 2023). To circumvent the complexities of RL, methods like Diffusion-DPO have adapted direct preference optimization techniques to efficiently align diffusion models with human preferences without an explicit reward model (Wallace et al., 2024). These lines of work, from inference-time steering to training-time alignment, are crucial for developing controllable and ethically aligned generative models.

## 6 CONCLUSION

We presented Reward-Guided Langevin Refinement, an inference-time framework that aligns deterministic ODE-based diffusion samplers to arbitrary differentiable rewards without updating model parameters. Our key idea is to interleave, at each step, a short Langevin refinement targeting a *pulled-back* energy tilt and a deterministic denoising push. We provided two simple, composable guarantees: (i) Langevin refinement samples from the layer-wise tilted law $p_t(x)\exp(\lambda_{t-1}r(f_t(x)))$; (ii) the deterministic push $x_{t-1} = f_t(x_t)$ transports this tilt exactly to $p_{t-1}(x)\exp(\lambda_{t-1}r(x))$. By iterating across steps with a decreasing reward schedule, the output distribution realizes the DPO-style exponential tilt, achieving preference alignment entirely at inference time.

Empirically, our methods consistently increases the expected reward across diverse datasets and reward models (Aesthetic, NIMA, and classifier-based class guidance), while preserving fidelity under matched wall-clock. Notably, *two* Langevin steps already deliver clear gains and *twenty* steps yield pronounced improvements, revealing a favorable reward–efficiency trade-off.

**Limitations** Our method introduces extra inference-time computation because of the Langevin dynamics. When more Langevin steps are used, it will definitely increase the expected reward of generated pictures, but will also lead to longer generation time. Also, the generated pictures would inherit the biases of the chosen reward.

**Future Work.** Beyond alignment at inference time, our framework may serve as a diagnostic lens for reward functions themselves. Because our method climbs reward gradients while preserving a principled diffusion denoising steps, it provides a controlled way to probe whether a reward can be *hacked*—i.e., whether large reward gains can be achieved with minimal semantic improvement or even undesirable artifacts. Perhaps our method could also be used as a general tool for red-teaming reward models under responsible-use protocols, helping designers refine reward definitions, improve calibration, and reduce susceptibility to reward hacking.

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
