# OpenReview forum: "Inference-Time Reward-Guided Langevin Refinement for Diffusion Models"
_ICLR.cc/2026/Conference — ICLR 2026 Conference Withdrawn Submission_

### Official Review · Reviewer_cXh3 · 2025-10-29

**Soundness:** 2
**Presentation:** 2
**Contribution:** 2
**Rating:** 2
**Confidence:** 4

**Summary:**

This paper proposes a training-free Langevin refinement method suitable for the inference stage, which aligns ODE-based diffusion samplers with arbitrary differentiable reward functions; the method enables preference or target-guided generation during sampling by alternately using standard deterministic denoising update steps and Langevin reward-guided correction steps, and the authors prove in the paper that the proposed method can gradually recover the DPO-style exponentially tilted target distribution. Furthermore, results on multiple image datasets and reward models demonstrate consistent improvements, and qualitative results confirm that using only a classifier as a reward can effectively guide image generation toward specific categories.

**Strengths:**

1. The proposed framework introduces a training-free, inference-time reward-guided refinement mechanism. It is compatible with any standard diffusion ODE solver and differentiable reward function, thus featuring broad applicability.
2. The Reward-Guided Langevin Refinement method presented in the paper does not rely on specific pre-trained models. It can also be combined with different reward models, which is conducive to further exploring the performance of Reward-Guided Langevin Refinement.
3. The effectiveness of the method proposed in the paper is mathematically proven. The final iterative results conform to the optimal output distribution in the form of Direct Preference Optimization (DPO).

**Weaknesses:**

1. As mentioned in the paper, generated images inherit the inherent biases of the selected reward model. If the reward model has issues such as data bias, labeling errors, or unreasonable design during training, the generated images will also be difficult to meet expectations.
2. The proposed method only discusses improvements in unconditional generation, while improvements in conditional generation with guidance are not addressed.
3. Although the method is training-free, it requires adding multiple Langevin refinement steps after each denoising step (up to 20 steps per denoising step). Each Langevin refinement step needs to call a pre-trained diffusion model and a pre-trained reward model once, resulting in additional computational costs under the same number of denoising steps.
4. The experimental results only provide improvements compared to the baseline, but do not include a performance comparison with other training-free methods under the same inference resources.
5. The experiments do not report standard generative model metrics (such as FID, precision/recall, or coverage) to evaluate whether the introduced reward gain comes at the cost of reduced perceived quality or diversity.

**Questions:**

1. Was a large model used as an aid in the implementation process of the paper? Why is the use of large models not introduced in the chapters?
2. From Formula (3.9) and Algorithm 1, each Langevin refinement step requires calling a pre-trained diffusion model and a pre-trained reward model once. Can you provide a comparison of results between using Langevin refinement and not using Langevin refinement under the same computational cost?
3. Can you provide more complete experimental results to show the improvement in FID, precision/recall, or coverage of generated images?

---

### Official Review · Reviewer_x1WF · 2025-10-31

**Soundness:** 2
**Presentation:** 2
**Contribution:** 2
**Rating:** 2
**Confidence:** 3

**Summary:**

The paper proposes an inference-time training free alignment approach to differentiable rewards for ODE samplers.

**Strengths:**

- The paper is reasonably well-written and has a coherent narrative.
- Mathematical problem formulations and derivations can be followed and strengthen the arguments.

**Weaknesses:**

- The experimental results are pretty limited. Both quantitative and qualitative. There are numerous settings proposed lately (less of benchmarks though) for alignment and guidance of diffusion models. This can help strengthen the claims and demonstrate the efficacy of the proposed approach. There is no comparison against related state-of-the-art baselines. There are a ton of inference-time alignment/guidance approaches out there.
- Increase in rewards does not always imply meaningful improvement in quality, this is when reward hacking can kick in. There is little discussion around that. I would consider pareto-front plots illustrating rewards vs divergence. I'll share a few reference below for inspiration.
- I wonder how the authors would handle non-differentiable rewards, for examples for compressibility. Maybe this can be inspired by the approach proposed in sampling based inference-time alignment approaches.

Below references regarding the remarks above:
- Sampling approach: https://arxiv.org/pdf/2409.15761
- Sampling approach for non-differentiable rewards: https://arxiv.org/abs/2502.00968
- Generic for differentiable rewards: https://arxiv.org/abs/2302.07121

**Questions:**

- Please proof-read the paper thoroughly once again, consistency across the draft helps. Why Limitations ... and Future Works"." ... Some headings end with dot and others don't. Some all words start with caps, the other don't.

- I'm not sure what Figure 2 even means. What is the impact of the initial image ... are you here referring to a Text+Image to Image generation? This needs to be elaborated.

---

### Official Review · Reviewer_7uqM · 2025-11-01

**Soundness:** 2
**Presentation:** 2
**Contribution:** 2
**Rating:** 4
**Confidence:** 3

**Summary:**

This paper proposes a training-free, inference-time method for aligning pretrained diffusion models with arbitrary differentiable reward functions. The method interleave standard deterministic ODE-based denoising steps with short Langevin refinement steps that target a "pulled-back" energy tilt derived from Direct Preference Optimization principles. The method is plug-and-play and requires no additional training or data, and it works with various reward models. Theoretical guarantees establish that the procedure yields the intended exponential-tilted distribution at each denoising step, culminating in the DPO-style optimal distribution at the output. Empirical results across multiple datasets (CelebA-HQ, AFHQ, LSUN Bedroom, ImageNet) and reward functions demonstrate consistent increases in expected reward with more Langevin steps.

**Strengths:**

1. Theoretical rigor: This paper provides a clean theoretical framework with proofs showing the method achieves the intended DPO-style distribution through compositional guarantees at each layer.
2. Plug-and-play flexibility: The method proposed in this paper works with any pretrained ODE-based sampler and arbitrary differentiable rewards without retraining, making it highly practical.

**Weaknesses:**

1. Limited experimental analysis: Primarily evaluates using the reward of generating images without detailed quantitative and qualitity analysis of generated samples. This makes it difficult to assess whether reward improvements correspond to qualitative improvements. And there is no comparison with existing inference-time guidance methods.  Can you provide more comparison results with baseline methods on the generated quality?
2. Insufficient utilization of 9-page limit: This paper is only 8 pages of content, leaving substantial space unused. This missed opportunity could have been used for more details for the introduction of the overall algorithm and analysis of the main results.
3. Missing discussion on the trade-off between efficiency and quality: While the paper acknowledges increased inference time as a limitation, it is necessary to provide more experiments to discuss the trade-off between the reward and Langevin steps. Can you add experiments to discuss this?
4. Lack of introduction to backbone: Sec. 4.1 merely states "We use a pretrained diffusion backbone with a deterministic ODE-based sampler (e.g., DDIM/EDM probability-flow)" without specifying which actual model architecture. Can you provide more details on the model backbone?
5. Limited task diversity: Sec. 4.3 introduce the class guidance image generation with your method. Can the method integrate with other task, such as a more widely-used task text-to-image guidance generation? Can you include more discussion about different tasks?

**Questions:**

In weakness.

---

### Official Review · Reviewer_ufzN · 2025-11-04

**Soundness:** 3
**Presentation:** 2
**Contribution:** 1
**Rating:** 2
**Confidence:** 4

**Summary:**

The paper proposes a training-free method that interleaves reward-guided Langevin refinement with deterministic ODE denoising to align diffusion models to differentiable rewards. It proves that each step realizes an exponential tilt of the layer distribution and that iterating recovers a DPO-style energy-based target at the output. Across many datasets and rewards, it consistently increases rewards during guidance.

**Strengths:**

- Theory with guarantees: layer-wise exponential tilt and a formal connection to DPO ensuring correctness
- Practicality: training-free, plug-and-play with standard ODE samplers, architecture-agnostic, and compatible with arbitrary differentiable rewards
- Empirical strength: consistent reward gains across datasets/rewards and effective class guidance at inference time without parameter updates

**Weaknesses:**

Weak Novelty / Contribution
- Several important related works are missing (refer to [1-4], but more are out there).
- Many of these prior studies have proposed inference-time guidance methods for T2I diffusion models, and some have even introduced generalizable frameworks.

Limited Empirical Setup
- Please specify the exact pretrained diffusion backbone used in Sec. 4.1. Based on the quality in Fig. 2, the experiments seem to lack results on more recent backbones.
- Comparisons with recent baselines are missing [1–3].
- The reward models used in the experiment are overly simple. Please consider more diverse and non-differentiable rewards (see [1]).

---
[1] Test-time alignment of diffusion models without reward over-optimization, ICLR, 2025

[2] Inference-time scaling for diffusion models beyond scaling denoising steps, CVPR, 2025

[3] A General Framework for Inference-time Scaling and Steering of Diffusion Models, ICML, 2025

[4] Inference-Time Alignment in Diffusion Models with Reward-Guided Generation: Tutorial and Review, ArXiv, 2025

**Questions:**

See weaknesses above

---

### Note · Authors · 2026-01-24

**Comment:**

We choose to withdraw this paper from ICLR 2026.

**Withdrawal Confirmation:**

I have read and agree with the venue's withdrawal policy on behalf of myself and my co-authors.